# Translating Senotherapeutic Interventions into the Clinic with Emerging Proteomic Technologies

**DOI:** 10.3390/biology12101301

**Published:** 2023-10-02

**Authors:** Amit K. Dey, Reema Banarjee, Mozhgan Boroumand, Delaney V. Rutherford, Quinn Strassheim, Thedoe Nyunt, Bradley Olinger, Nathan Basisty

**Affiliations:** Translational Geroproteomics Unit, Translational Gerontology Branch, National Institute on Aging, National Institutes of Health, Baltimore, MD 21224, USA; amit.dey2@nih.gov (A.K.D.); reema.banarjee@nih.gov (R.B.); mozhgan.boroumand@nih.gov (M.B.); delaney.rutherford@nih.gov (D.V.R.); quinn.strassheim@nih.gov (Q.S.); thedoe.nyunt@nih.gov (T.N.); bradley.olinger@nih.gov (B.O.)

**Keywords:** senescence, aging, mass spectrometry, proteomics, biomarkers, senolytics, senomorphics, surfaceome, SASP, senotherapeutics, CETSA, TPP, drug discovery, geroscience

## Abstract

**Simple Summary:**

The accumulation of senescent cells is now widely known to be a driver of aging and many age-related pathologies, such as neurodegeneration and type 2 diabetes, among others. Targeting senescent cells for selective removal or altering the proteins they release are promising therapeutic strategies against age-related diseases. However, the biology of senescent cells is complex, dynamic, and heterogeneous. In order to better identify pathology-driving senescent cells and develop therapies to alter their complex biology or drive them toward cell death, a detailed and comprehensive understanding of senescence-associated proteins and the mechanisms that enable senescent cells to evade cell death is required. Major developments in proteomic workflows over the past decade have enabled an increasingly comprehensive, quantitative, and specific molecular profiling and interrogation of cellular mechanisms. In this review, we discuss the current state of translational senescence research and how modern proteomic technologies, particularly mass spectrometry-based proteomics, can accelerate our understanding of the fundamental mechanisms that drive senescence and robustly probe the proteomic phenotypes of heterogenous populations of senescent cells. We will focus on how these fundamental biological insights will ultimately accelerate the development of senescence-targeting therapies, or senotherapeutics.

**Abstract:**

Cellular senescence is a state of irreversible growth arrest with profound phenotypic changes, including the senescence-associated secretory phenotype (SASP). Senescent cell accumulation contributes to aging and many pathologies including chronic inflammation, type 2 diabetes, cancer, and neurodegeneration. Targeted removal of senescent cells in preclinical models promotes health and longevity, suggesting that the selective elimination of senescent cells is a promising therapeutic approach for mitigating a myriad of age-related pathologies in humans. However, moving senescence-targeting drugs (senotherapeutics) into the clinic will require therapeutic targets and biomarkers, fueled by an improved understanding of the complex and dynamic biology of senescent cell populations and their molecular profiles, as well as the mechanisms underlying the emergence and maintenance of senescence cells and the SASP. Advances in mass spectrometry-based proteomic technologies and workflows have the potential to address these needs. Here, we review the state of translational senescence research and how proteomic approaches have added to our knowledge of senescence biology to date. Further, we lay out a roadmap from fundamental biological discovery to the clinical translation of senotherapeutic approaches through the development and application of emerging proteomic technologies, including targeted and untargeted proteomic approaches, bottom-up and top-down methods, stability proteomics, and surfaceomics. These technologies are integral for probing the cellular composition and dynamics of senescent cells and, ultimately, the development of senotype-specific biomarkers and senotherapeutics (senolytics and senomorphics). This review aims to highlight emerging areas and applications of proteomics that will aid in exploring new senescent cell biology and the future translation of senotherapeutics.

## 1. Introduction

Aging comprises a cascade of underlying cellular and molecular processes, commonly referred to as the hallmarks of aging, that lead to a gradual loss of function and increased susceptibility to diseases. One of the most studied hallmarks of aging, cellular senescence, is a key driver of aging and age-related diseases [1]. Cellular senescence is a complex stress response resulting from a variety of sub-lethal stresses that permanently alter the state of a cell. Three of the defining features of senescence are a permanent arrest of cell proliferation, an increased secretion of a variety of bioactive molecules known as the senescence-associated secretory phenotype (SASP), and a resistance to apoptosis [2,3]. Despite an arrest of cell growth, senescent cells remain metabolically active and secrete a robust SASP that comprises bioactive molecules, including metabolites, proteins, and lipids [4]. The chronic presence of senescent cells and the SASP are connected to various age-related disorders such as cancer, diabetes, neurodegeneration, osteoarthritis, and cardiovascular disease [5,6,7]. Therefore, the elimination of senescent cells and the SASP are promising approaches for combating age-related diseases and improving healthspan [8].

Causal linkage among aging, cellular senescence, and age-related diseases has caused the emergence of ‘senotherapeutics’ [8], a catch-all term that refers to therapeutic interventions that target senescent cells. Two common classes of pharmacological senotherapeutics include senolytics and senomorphics [9,10]. Senolytics are chemical compounds that selectively kill senescent cells. Non-pharmacological senotherapeutic approaches, such as vaccines or immunotherapies, are also proposed options for the selective elimination of senescent cells [11,12,13]. Interventions that reduce the upstream inducers of senescence, such as DNA damage, ROS, inflammation, or metabolic imbalance, likely confer senotherapeutic benefits by reducing the initiation of senescence. Additionally, targeting cell populations that drive senescence, such as aged immune cells [14], may reduce senescent cell burden. Senomorphics are the agents that can block or otherwise modulate the SASP to reduce its detrimental activity and mitigate aging phenotypes [15,16]. The discovery of the SASP and its potency as a driver of aging have greatly increased interest in its comprehensive characterization, both to explore new mechanisms of aging and identify new biomarkers that indicate ‘senescence burden’, a useful metric for the clinical translation of senotherapeutic approaches [1,17,18].

Protein biomarkers are essential due to their diagnostic, prognostic, and predictive power, as well as identifying and stratifying patients for treatment and measuring the efficacy of therapies. Mass spectrometry-based proteomics is a powerful, versatile, and robust technology used for comprehensively quantifying and discovering proteins with unrivaled specificity [19]. Given the robust and heterogeneous proteomic phenotypes associated with senescence and the SASP, the discovery and profiling of their proteomic signatures require the large-scale, unbiased, and quantitative abilities that mass spectrometry can provide. The presence of senescence-associated proteins in circulation in recent studies also suggests the use of proteomic technologies will be required for the detection and quantification of senescence biomarkers in blood. Numerous innovations in mass spectrometry workflows for biomarker and drug discovery have been made in recent years, opening new opportunities to accelerate the development of senotherapeutics.

Here, we review the available technologies for identifying, validating, and prioritizing protein biomarkers and therapeutic targets identified via mass spectrometry and how these technologies may be leveraged for the clinical translational of senotherapeutics. We describe the technological advancements that have enabled researchers to address challenges inherent to the proteomic analysis of blood, such as the wide dynamic range of protein concentrations, and discuss multiple workflows that can be leveraged for the discovery of senescence biomarkers, senolytic targets, and cell-surface proteins. We also highlight how modern mass spectrometry-based technologies will open the door for future clinical applications, develop translationally relevant approaches to quantify aging and cellular senescence, and develop therapeutics for enhancing human healthspan.

## 2. Targeting Senescent Cells with Senotherapeutics

Pharmacological compounds [20] and biologics, such as platelet-rich plasma [21], have been identified as candidate senotherapeutics for the treatment of aging and the pathologies of aging. Within this context, we highlight small-molecule senotherapeutic compounds that have demonstrated notable efficacy in preclinical studies, positioning them as promising candidates for future clinical investigation. Additionally, senotherapeutic drugs exhibit cell-type specificity [22,23], potentially resulting in varied effects on senescent cells originating from different tissues and senescence-inducing stimuli. Ultimately, comprehensive screening of these drugs across multiple subtypes of senescent cells is essential to determine their efficacy and specificity.

### 2.1. Natural Compounds: Flavonoids

Flavonoids are a diverse group of polyphenolic plant compounds isolated from several common fruits, vegetables, and herbs. Research suggests that certain flavonoids exhibit senotherapeutic properties via antioxidant effects [24], anti-inflammatory properties [24,25], direct senolytic activity [25,26,27], and overall reductions in the SASP [28,29,30]. Flavonoids potentially exhibit antiaging properties through one or a combination of many described effects on age-related, metabolic, and growth pathways. For example, fisetin and quercetin influence the 5’ adenosine monophosphate-activated protein kinase (AMPK) pathway [28,29] and the mitogen-activated protein kinase (MAPK) pathways [30,31,32], thereby modulating cell growth, metabolism, and inflammatory responses. Flavonoids can also modulate the phosphoinositide 3-kinase (PI3K) and protein kinase B (PKB/Akt) pathways [33,34,35], which regulate cell growth and survival. Finally, flavonoids have been shown to interact with senescence-associated pathways including the p53/p21 pathway [36,37,38] and the mechanistic target of rapamycin (mTOR) pathway [24,29,39], a key regulator of the inflammatory arm of the SASP [40]. Although studies have indicated the involvement of flavonoids in numerous pathways, the direct protein targets of flavonoids that are responsible for senolytic activity remain poorly understood and are an ongoing area of research. It is currently unclear whether flavonoids target each of the pathways mentioned above, or whether these observed effects reflect the downstream results of a primary action on a single aging mechanism target. Since the targets of flavonoids have not yet been identified, their mechanisms of action are worth further investigation.

### 2.2. Dasatinib + Quercetin

The combination of dasatinib and quercetin is one of the most potent and highly studied senolytic treatments. Primarily used in the treatment of certain types of cancers, dasatinib works as a tyrosine kinase inhibitor to regulate cell growth and division. Dasatinib has been shown to induce senescent cell apoptosis by inhibiting specific signaling pathways that promote survival in senescent cells, such as forkhead transcription factor (FOXO)-p53 and PI3Kδ [41,42]. Additionally, dasatinib can target and inhibit serpin family E member 1 (SERPINE1), also commonly referred to as plasminogen activator inhibitor 1 (PAI-1) [43,44], which is a family of proteins involved with blood clotting, tissue remodeling, cell migration, and known components of the SASP. By targeting SERPINE1, dasatinib may also help regulate processes related to cancer progression [45,46,47], but the exact mechanism by which dasatinib acts on PAI-1 remains unclear. Finally, dasatinib has been shown to reduce reactive oxygen species (ROS) [48,49]. Quercetin, a naturally occurring flavonoid, possesses some unknown cellular targets and interacts with several pathways associated with cellular growth and survival (see Flavonoids section). Several preclinical studies have explored how the combination of these two drugs enables the clearance of senescent cells [20,50], reduces several markers of senescence [51], and allows for the amelioration of age-related pathologies in animal models [52].

### 2.3. Metformin

Commonly used in the treatment of type 2 diabetes, metformin has gained popularity as a potential antiaging intervention following a human study that found diabetic patients treated with metformin outlived age-matched non-diabetic control patients [53]. Metformin has since been widely studied for its potential longevity benefits [54,55]. One study found a decrease in senescent cell number in mouse olfactory ensheathing cells treated with metformin by examining SA- β gal activity. This study also examined SASP activity and found that metformin treatment interfered with the pro-inflammatory nuclear factor kappa light chain enhancer of activated B cells (NF-kB) [52]. Additionally, metformin has been shown to inhibit the mTOR pathway [54,56,57], a known regulator of the inflammatory SASP [40]. Metformin is also able to mitigate chronic inflammation via senomorphic modulation of inflammatory signaling pathways [55,58,59] such as NF-kB and Janus kinase–signal transducer and activator of transcription (JAK-STAT) to reduce the production of pro-inflammatory cytokines.

### 2.4. Rapamycin

Originally discovered as an antifungal agent and later found to have immunosuppressive and anticancer properties, rapamycin is the most robust pharmacological aging intervention to date in mice, conferring large healthspan and lifespan increases across multiple studies [60,61,62,63]. While there are multiple proposed mechanisms by which rapamycin slows aging, one key proposed mechanism is senotherapeutic activity [64,65]. Rapamycin primarily interacts with the mTOR complexes, which regulate cell growth and metabolism [66]. By inhibiting the mTOR pathway, rapamycin may modulate these cellular processes and even reduce the appearance of new senescent cells by promoting homeostasis via the removal of dysfunctional proteins and aged organelles through autophagy. Rapamycin has also been shown to reduce the SASP [67,68,69] to inhibit the production of pro-inflammatory molecules and their signaling pathways. Through an mTOR-dependent reduction in NF-κB transcriptional activity, rapamycin can inhibit the translation of inflammatory SASP factors [40]. The levels of SASP components is also regulated via eukaryotic initiation factor 4E-binding protein 1 (4EBP1)-mediated translation of MAPK- activated protein kinase 2 (MAPKAPK2 or MK2), phosphorylating and inhibiting the ability of the RNA-binding protein ZFP36L1 to degrade SASP transcripts [70]. The use of rapamycin and its analogs as senotherapeutics is an active area of research, and current studies aim to optimize dosages and treatment duration to explore their efficacy and safety.

### 2.5. Nutlin

Nutlin is a class of protein–protein inhibitors that are known to inhibit the MDM2 proto-oncogene (MDM2) protein interaction with the p53 transcription factor [71]. This pathway is involved in apoptosis and therefore nutlin has been used as a cancer treatment to cause apoptosis in cancer cells. Other studies investigated this same pathway and its effects on senescence, specifically with nutlin-3a. One study induced senescence in the retinal pigment epithelium (RPE) to represent age-related macular degeneration (AMD) in vitro and treated the cells with nutlin-3a [72]. They found that nutlin-3a induced cell death in 50% of the senescent RPE cells while the non-senescent cells did not have significant cell death. The same study also tested the effects of nutlin-3a in vivo in mouse models, finding that treatment ameliorated the degeneration of the mouse retina [72]. Ongoing research continues to study the senolytic effects of nutlin-3a, such as treatment with senescent melanoma cells [73]. Nutlin remains a possible choice for the treatment of senescence, especially in epithelial and melanoma cells.

### 2.6. Bcl-2 Family Member Inhibitors

B-cell lymphoma 2 (BCL-2) encodes the protein family that regulates cell death by inhibiting or inducing apoptosis [74]. In senescence, the BCL-2 family of proteins is a ‘pro-survival’ pathway that confers resistance to apoptosis to many senescent cell types [75], and most current senolytics target this pathway. Therefore, studies have sought drugs that can inhibit this class of proteins causing apoptosis in senescent cells [75,76]. ABT-263, or navitoclax, is a drug that selectively kills senescent fibroblasts (IMR90), renal epithelial cells, human umbilical vein epithelial cells (HUVECs), and mouse embryo fibroblasts [75,76]. This drug also cleared senescent bone marrow stem cells and senescent muscle cells in sub-lethally irradiated P16-3MR transgenic mice [75]. ABT-737 is a similar inhibitor to navitoclax and has higher potency with fewer of the negative side effects [77]. The successful elimination of senescent skin cells in mice has also been accomplished with ABT-737 [77]. Finally, A1331852 and A1155463 are two related inhibitors which reduced the survival of senescent HUVECs and IMR90s in vitro [78]. These drugs inhibit the BCL-2 family, causing apoptosis in senescent cells, but are only successful in certain cell types [78]. Single or combination treatments with BCL-2 inhibitors may be a promising treatment for the specific removal of senescent cells in humans.

### 2.7. Hsp90 Inhibitors

Heat shock protein 90 (HSP90) acts as a molecular chaperone to stabilize misfolded or unfolded proteins under cellular stress [79]. Previous studies have found that HSP90 promotes cell survival via an interaction and stabilization of factors such as AKT or extracellular signal-regulated kinase (ERK), and an inhibition of their downstream signaling pathways, including Foxo3a, mTOR, and others [80]. Senescent cells contain elevated levels of p-AKT, the activated form of AKT, which is stabilized by HSP90 [81]. The disruption of this Interaction by HSP90 inhibitors reduces p-AKT, resulting in senolysis. In one study, the HSP90 inhibitors geldanamycin and 17-AAG effectively killed multiple senescent cell types in two species, including murine embryonic fibroblasts and human lung fibroblasts (IMR90 and WI-38). Futher, these treatments were successful in senescent cells produced by different senescence-induction treatments, including oxidative stress, genotoxic stress, and telomere shortening [81]. The same study demonstrated the effectiveness of the inhibitors in vivo in mice lacking Ercc1 (Ercc1 −/Δ), a progeroid mouse model, by reducing senescent cell burden and delaying multiple aging phenotypes [81]. The results of these studies reveal an interesting future for HSP90 inhibitors as selective senolytics that could be further investigated as a possible treatment for aging-related pathologies.

## 3. Development of Senescence-Associated Biomarkers and Therapeutic Targets for Clinical Translation

The pipeline for the translation of senotherapeutics and biomarkers into the clinic is a multipronged approach that can be accelerated by mass spectrometry-based proteomic technologies (Figure 1). This pipeline encompasses a workflow beginning with the generation of biomarkers and target candidates in cell culture models to the validation of biomarkers in human tissues and cohorts, often aided by technological developments in proteomics. Since its original identification with proteomic arrays in 2009 [82,83], the senescence-associated secretory phenotype (SASP) has captured the attention of many as a source of biomarkers that can assess senescent cell burden. The SASP has since been shown to be heterogeneous, dependent on cell type, duration of senescence, and the method of senescence induction [1,84]. The release of molecules enclosed in extracellular vesicles is a more recently described component of the SASP that exerts bioactive effects, including paracrine senescence [85], and contributes to disease pathologies such as osteoarthritis [86]. Due to the dynamic nature of the senescent secretome, compilations of SASP proteome profiles in different cell types have been generated using data-independent acquisition mass spectrometry and other proteomic approaches, such as the ‘SASP Atlas’ [2]. Studies have also identified ‘core’ SASP factors that characterize the senescent proteome irrespective of the method of induction, such as in lung fibroblasts [2], a subset of which were significantly associated with aging or aging-related outcomes in human studies [6,87]. The goal continues to be to fully characterize the SASP and other senescence-associated proteins, both the core factors and those specific to individual cell lineages, along with their clinical associations. Based on these insights, clinical researchers are now targeting senescence via senolytic drug administration in diverse patient populations, to gauge its potential clinical benefits. In this section, we review the recent literature on senescence-associated biomarkers, their clinical associations, and outcomes of senolytic clinical trials.

### 3.1. SASP in Circulating Plasma

Characterizing the levels and roles of the SASP and senescence-associated molecules in humans during aging, disease states, and aging-associated outcomes (mortality, multimorbidity, frailty, etc.) is critical for elucidating the role that SASP factors play in driving the pathologies of aging. The discovery of senescence-associated protein associations with age-related outcomes is also critical for identifying important clinical outcomes for future senotherapeutic trials. Proteomic approaches have been instrumental thus far in establishing the role of the SASP in humans by measurement of SASP in circulating plasma and examining its associations with aging and age-related phenotypes [6]. Indeed, studies of blood samples from human cohorts have identified clinical associations with the SASP. For example, Schafer et al. reported that among their 19 age-associated SASP factors, growth differentiation factor-15 (GDF15), Osteopontin (OPN), and TNF receptor 1 (TNFR1) were positively associated with frailty after adjustment for age, body mass index (BMI), and sex [88]. Fielding et al. found 13 significant inverse correlations between short physical performance battery (SPPB) scores and circulating SASP levels, with activin A, vascular endothelial growth factor A (VEGFA), and interleukin 15 (IL15) among the most strongly correlated [89]. Furthermore, the assessment of senescence burden is important not only for its correlation with age-related phenotypes but also for its ability to predict the onset of such conditions. Machine learning modeling has shown that accounting for circulating SASP concentrations enhances the prediction of conditions like mobility disability, compared with covariates like age, sex, race, and BMI [89].

SASP levels in plasma have also been associated with health outcomes after the onset of diseases such as cancer, cardiovascular disease, and respiratory infections. Plasma levels of SASP proteins, including GDF15, TNFR1, and others, are positively associated with complications in patients following surgery for cardiovascular disease. Similarly, TNF receptor superfamily member 6 (FAS) and OPN are correlated with postoperative complications in ovarian cancer patients [88]. Patients suffering from airway-related diseases such as chronic obstructive pulmonary disease (COPD) who died after infection with COVID-19 had elevated levels of macrophage chemotactic protein (CCL25) and other SASP proteins in plasma, as well as other SASP markers, such as IL6 and TGF-B, elevated in the lung tissue [90]. Serum levels of the senescence-associated protein PAI1 were found to be associated with greater impairment among patients with airway-related diseases such as COPD [91]. Plasma levels of the senescence marker insulin-like growth factor binding protein 7 (IGFBP7) were associated with worse health outcomes, such as hospitalization, in a cohort of patients suffering from heart failure [92]. Mental health outcomes in older adults have also been associated with blood plasma levels of other SASP factors, such as IGFBP6, CCL4, and TNF, among others, when cumulatively considered as a composite index (22 SASP factors in total) [93].

### 3.2. Senescence-Associated Markers in Tissues and Biofluids

Beyond the SASP in circulation, the presence of senescence signatures in tissues and biofluids has been associated with aging-related clinical outcomes. For instance, a study by Rouault et al. showed that in adipose biopsies obtained from obese patients, samples that expressed higher levels of senescence-associated β-galactosidase activity had up to a twofold increase in the secretion of factors like leptin, PAI1, IGFBP3, and CCL2 [94]. A higher incidence of type 2 diabetes was also found among patients with samples exhibiting this senescent profile [94]. In urine samples, Yu et al. found 23 differentially expressed secreted proteins from healthy aging and non-healthy aging (e.g., diagnoses of cancer, cardiovascular and pulmonary diseases, and neurodegenerative disorders) cohorts [95]. These markers include cathepsin D, afamin, matrix remodeling-associated protein 8, and angiopoietin-related protein 2, spanning biological processes related to senescence, vascular dysfunction, extracellular matrix remodeling, and inflammation [95]. Looking at senescent glial cell markers in CSF, Dai et al. found that HGF, MIF, and TSP2 were upregulated with age in patients with Alzheimer’s disease (AD) pathology [96]. They also found that accounting for YKL-40, serpinA3, and TSP2 could help discriminate patients with AD and non-AD cognitive disorders from patients with normal cognition [96]. Overall, the small number of studies examining senescence markers in non-circulating biofluids and tissues highlight the potential of these specimens for novel biomarker discovery and development.

### 3.3. Senolytic Clinical Trials

With a comprehensive understanding of the signs and symptoms of senescent cell burden, clinical studies are looking to target senescent cells through senolytic administration to alleviate disease phenotypes. Mouse models gave the first successful indications that senolytic treatments like dasatinib plus quercetin or fisetin could promote healthy aging [26,42]. Consequently, the first human clinical trials of dasatinib plus quercetin (D+Q) treatment for patients with idiopathic pulmonary fibrosis (IPF) and diabetic kidney disease (DKD) showed preliminary evidence of improved physical function measures, decreased adipose tissue senescent cell burden, and decreased circulating SASP factors like IL-1α and IL-6 [50,97].

Recently reported senolytic trial findings have deepened our understanding of how assessing and targeting senescence in humans can translate into clinical settings. Nambiar et al. conducted a randomized placebo-controlled pilot study of senolytic treatment for IPF as a follow-up to their open-label pilot study [98]. The feasibility and tolerability of D+Q were confirmed, with no serious adverse events in the senolytic treatment group, but a threefold increase in non-serious adverse events (e.g., feeling unwell, nausea, fatigue) compared to the placebo treatment group [98]. Looking at potential senescence biomarkers, supplemental data indicated that in collected blood samples, senolytic treatment resulted in a smaller increase in the inflammation marker C-reactive protein (CRP), and a decrease in the senescence marker glycoprotein non-metastatic melanoma protein B (GPNMB), compared to placebo [98]. Zhu et al.’s study investigated other senescence protein biomarkers like α-Klotho in urine samples from D+Q-treated IPF patients [99]. As a ‘geroprotective’ protein inversely related to cellular senescence, α-Klotho was found to increase in urine following D + Q treatment compared to before treatment, demonstrating the therapeutic utility of senolytic treatment as well as the plausibility of tracking urinary α-Klotho as a measure of senescence burden [99]. α-Klotho also inversely correlated with urinary markers that increase with senescence, such as IL-6, MCP-1, TIMP1/2, and MMP7/8, pointing to broader panels that may help assess the effectiveness of senolytic treatment [99]. D + Q treatment was also investigated as a treatment for patients with AD in the open-label SToMP-AD clinical trial [100]. This study proposed to investigate the penetrance of D + Q in CSF and blood in patients with early symptomatic AD and assess resultant changes in cognitive and physical outcomes and SASP levels [100]. Preprinted results of this study have found that D + Q was increased in the blood, and D but not Q was detected in CSF [101]. Although most cognitive outcomes did not show significant changes, SASP markers in the blood (e.g., IL-17E, IL-21, IL-23, and VEGF) and in the CSF (TARC, IL-17A, I-TAC, Eotaxin-2, Eotaxin, and MIP-1α) showed significant decreases following treatment [101]. Additionally, UNITY Biotechnology, a startup that develops therapeutics against aging with a focus on senescence, has been conducting clinical trials of senolytic Bcl-xL inhibitor UBX1325 in patients with diabetic macular edema (DME) and wet age-related macular degeneration (wet AMD) [102,103]. The results from their phase 1 study showed clinical improvement, with increases in best-corrected visual acuity (BCVA) and decreases in central subfield thickness (CST) that were sustained for between 3 and 6 months following a single intravitreal injection of UBX1325 [102,103]. Unity’s phase 2 randomized, double-masked, sham-controlled study has announced 24-week data, including overall drug safety and tolerability, absence of severe adverse effects, and similar clinical improvements with a mean 7.6-letter gain in BCVA and a mean 5.4 decrease in CST [104,105]. Collectively, these clinical results point to specific biomarkers of senescence that are readily detectable in patient samples and provide a way to correlate a decrease in senescence burden with an amelioration of clinical outcomes.

Newly completed and ongoing senolytic trials highlight important clinical goals of assessing and targeting senescence in a variety of age-related pathologies. Phase 2 of the SToMP-AD clinical trial will evaluate the presence of 10 primary SASP factors in blood along with senescent T cell levels and other cognitive and behavioural improvements [106]. D+Q and fisetin trials recruiting survivors of hematopoietic stem cell transplants and childhood cancer will be looking at physical health outcomes like frailty and walking speed while investigating measures of senescence-like p16 mRNA levels from CD3^+^ T-cells [107,108]. More studies on D+Q and fisetin are examining the role of senescence in musculoskeletal phenotypes, like osteoporosis, osteoarthritis, and frailty in elderly women, by conducting physical functionality tests and assessing biochemical signatures related to bone resorption and formation, OA-related articular cartilage degeneration, or inflammation, respectively [109,110,111]. A final trend is a group of studies looking at fisetin’s ability to improve outcomes for COVID-19 patients in multiple inpatient/outpatient settings, one of which will also be measuring senescent cell abundance and inflammation markers [112,113,114]. As we look forward to their reported findings, we see that biomarkers of senescence play a crucial role in relating senescent cell burden with the amelioration (or lack thereof) of these disease outcomes. While we discuss here the emerging use and need for proteomic biomarkers in clinical research, we direct readers elsewhere for more comprehensive reviews of current and planned senolytic trials [3,115].

Though not officially a senolytic, clinical researchers have also begun examining the utility of metformin as an ‘anti-aging’ and senotherapeutic drug. Notably, the Targeting Aging with MEtformin (TAME) trial has been planned as a collaboration between 14 institutions and 3000 elderly participants to examine whether metformin administration will delay the development or progression of various age-related diseases like cancer and dementia [116,117]. The TAME Biomarkers Workgroup established a short list of aging biomarkers to examine whether metformin treatment is modulating aging biology as expected. Multiple senescence-associated proteins will be measured including IL-6, TNFα-receptor I or II, GDF15, IGF1, and cystatin C [118]. Overall, the results of TAME and ongoing senolytic trials will be valuable in evaluating the utility of proteomic biomarkers of senescence and age-related outcomes in a clinical setting, as well as identifying potential clinical outcomes to be targeted with senotherapeutics.

## 4. Emerging Proteomic Technologies for Accelerating the Development of Senotherapeutics

In recent years, proteomics has become an attractive tool for uncovering senescence biology, biomarkers, and therapeutic targets. We anticipate the application of mass spectrometry (MS)-based proteomics will accelerate the translation therapies targeting senescent cells. MS-based proteomics is a gold-standard technique for large-scale protein identification and quantification in complex samples. To overcome the challenges associated with analyzing complex protein samples with a large dynamic range of protein concentrations, sophisticated separation techniques and advanced MS instrumentation have been developed to extend coverage and sensitivity to accommodate the detection of biologically relevant and low-abundance proteins. Here, we review current approaches in LC-MS-based proteomics, including data acquisition techniques, sample preparation, and methods for protein quantification. Further, we describe innovative applications of MS-based workflows that will facilitate the discovery of senescence biomarkers and therapeutic targets (Figure 2). Target-discovery approaches range from profiling of protein–ligand interactions to the characterization of the cell-surface proteome. Finally, we describe biomarker discovery approaches that focus on the identification of low-abundance and highly specific proteoforms and PTM-level biomarkers that are challenging to detect in blood using traditional untargeted proteomic approaches.

### 4.1. Emerging MS-Based Approaches for Biomarker Discovery

#### 4.1.1. Data Acquisition and MS Instrumentation

MS data acquisition methods and analysis pipelines have significantly improved data reproducibility, quantification, and completeness in mass spectrometry-based workflows. The advancement of data-independent acquisition (DIA or SWATH) methods is a major step toward increased reproducibility and quantitative accuracy for biomarker discovery compared with traditional data-dependent acquisition (DDA) methods applied in untargeted proteomics studies [4,119,120]. DIA MS acquires MS/MS spectra for all peptides in a sample without requiring precursor information, which overcomes several challenges associated with DDA, including reduced bias toward abundant peptide selection that results in data missingness and decreased quantitative interference at the MS1 level [119,121]. DIA-MS also offers a reliable and consistent analysis of complex proteomic samples [122,123] that have been validated in consortium-level biomarker studies [120]. DIA analysis also lends itself nicely to PTM profiling and quantification, as quantification at the MS2 level allows one to localize PTM sites and distinguish peptide isomers via site-localized fragment ions [124,125]. DIA-MS acquisition generally uses wider precursor isolation windows to activate all ions for collision in each mass-to-charge (*m*/*z*) range, repeated across the full mass range in a single duty cycle. Innovations to the size and placement of precursor isolation windows, such as variable-window DIA [126] and overlapped (staggered)-window DIA [127,128], were developed to improve peptide selectivity. The generation of the first comprehensive SASP profiles in SASP Atlas were generated utilizing variable-window DIA to identify candidate senescence biomarkers such as GDF15, STC1, and SERPINs that are associated with aging in human plasma [2]. Another newly developed novel acquisition method called ‘parallel accumulation–serial fragmentation’ (PASEF) [129], combined with DIA (diaPASEF) [130], improves sensitivity and scanning speed. Unlike typical DIA approaches based predominantly on chromatographic separation, diaPASEF increases the sensitivity and coverage of proteomic analysis with the inclusion of an additional ‘ion mobility’ peptide separation dimension.

#### 4.1.2. Identification of ‘Proteoform’-Level Biomarkers with Top-Down Proteomics

Hundreds of thousands of proteins are produced by approximately 20,000 human coding genes [18,131,132]. Following protein biosynthesis, post-translational modification (PTM) and proteolytic processing events can modify protein structures and interactions [133,134]. PTMs include chemical groups such as phosphorylation and acetylation and more complex glycan structures, among many others [135,136]. Any protein variation (protein products of a single gene) due to PTMs, alternative RNA splicing, and genetic mutation is defined by a term named ‘proteoform’ [137], which can have distinct functions and change dynamically in response to diverse stimuli [138,139]. Proteoforms are challenging to characterize because of their similar sequences and masses, which make proteomic analysis more complex [18]. Most ‘discovery’ proteomic approaches rely on peptide-level measurements or antibody binding at a specific location on a protein, and therefore are unable to distinguish proteoforms. However, proteoform-level measurements will potentially improve the sensitivity and specificity of proteomic biomarkers.

Top-down proteomics is a powerful technology for analysis of the intact proteins/peptides via tandem mass spectrometry to characterize unique proteoforms and localize PTMs [138,139,140] to reveal structural and functional information on each proteoform [141,142]. Based on recent studies, protein modifications and diseases can be closely associated with each other [143]. For instance, the relationship between the occurrence and development of cancer and phosphorylation sites has been described [144,145,146]. The Kelleher Research Group at Northwestern University has reported 29,620 unique proteoforms which are expressed from human blood and bone marrow, accessible in the Blood Proteoform Atlas (BPA) (https://blood-proteoform-atlas.org (accessed on 24 August 2023)). The potential applications of BPA in clinical validation, such as the prostate-specific antigen isoform (IsoPSA) test in prostate cancer, have been noted [147] and there are many more proteoform associations yet to be explored.

Proteoforms are potential biomarkers that can be directly connected to multifaceted phenotypes [142], and thus the separation of intact proteins is necessary and one of the most significant challenges of top-down proteomics. Intact proteins are typically separated using reversed-phase liquid chromatography (RPLC) coupled with tandem mass spectrometry (MS/MS) analysis. Separation techniques have been developed for multidimensional separation platforms such as capillary zone electrophoresis (CZE), two-dimensional (2D) gel electrophoresis, and 2D-LC platform using high-pH RPLC and low-pH RPLC separation [148,149]. In polyacrylamide gel electrophoresis (PAGE), Passively Eluting Proteins from Polyacrylamide gels as Intact species for MS (‘PEPPI-MS’) is a recent advancement for clean separation and extraction of intact proteins [150]. Future advancements in MS instrumentation and data analysis tools are needed for pervasive application of top-down proteomics in biomedical research [146]. such as streamling and upgrading of database search tools. For example, the presence of unexpected modifications complicates database matching. Fortunately, there is a rapid growth of top-down proteomics [151], propelled by community groups such as the Consortium of Top-Down Proteomics (www.topdownproteomics.org (accessed on 24 August 2023)) and initiatives such as the Human Proteoform Project, which strive to map the human proteoform at proteoform resolution [141].

#### 4.1.3. Sample Preparation Workflows to Overcome Challenges in Blood Biomarker Discovery

Circulating biomarkers are promising indicators to quantify age-related declines and senescent cell burden [1,152]. The translation of therapies, like senolytics and senomorphics, will require biomarkers of senescence burden to identify and stratify individuals with elevated senescent cell burden and evaluate the efficacy of senolytic interventions. Interrogation of the blood proteome is key to establishing senescence biomarkers and subsequent senotherapeutics. However, the high dynamic range of protein abundance in plasma/serum, which spans over 10 orders of magnitude, poses a challenge in the detection of lower abundance components using mass spectrometry [153]. The 22 most abundant proteins in blood, including albumin, immunoglobulins, myoglobin, transferrin, and haptoglobin, together constitute 99% of the total protein content, whereas albumin itself comprises nearly 60% of the total serum proteome [154]. Low-abundant proteins having biomarker potential may be obscured from detection by high-abundance proteins. Thus, sample preparation protocols are critical for minimizing sample complexity. Various strategies therefore play a crucial role in reducing sample complexity before LC-MS/MS analysis, including depletion of high-abundance proteins, affinity enrichment of low-abundant proteins, and a variety of chromatographic [155] and electrophoretic fractionation techniques [156]. The depletion of highly abundant proteins, reviewed in detail elsewhere [157], is a viable strategy for reducing the dynamic range at the expense of increasing the number of samples to be analyzed. Immunodepleting highly abundant proteins is a routinely applied method with high specificity and reproducibility [157]. Antibody-based depletion of the top 7 or top 14 high-abundance plasma proteins showed 25% gains in proteome coverage [158]. However, these gains come at the expense of added processing steps, additional variability, and the cost of the depletion columns. Another method based on bead-bound hexapeptides known as combinatorial peptide ligand library (CPLL, commercially named Proteominer) reduces the dynamic range of protein concentrations [159]. CPLL enriches low-abundant proteins in a reproducible manner [160], and with greater performance compared to the other available depletion/enrichment approaches, yielding 68.6% higher protein identification compared to the untreated serum [161]. The impact of CPLL in serum/plasma biomarker discovery is excellently reviewed elsewhere [162]. In one notable study, CPLL increased plasma protein identifications by 22% in breast cancer patients, of which 23 were differentially expressed and thus potential biomarker candidates [163]. Another study applying CPLL identified 26 differentially expressed proteins, of which ficolin-2 was reported as a potential diagnostic marker for rheumatoid arthritis [164].

Each strategy employed for reducing the dynamic range of protein samples has its own merits and drawbacks [157,165] that impose trade-offs between sample throughput and proteome coverage. Extensive offline fractionation increases the number of samples and MS analysis time, whereas depletion/enrichment strategies add processing steps that introduce technical variability and cost. Further depth can be achieved by utilizing a combination of approaches [166,167]. A recent study assessed the plasma proteome by comparing depletion via Human 6 (Hu6) and Human 14 (Hu14) columns and the ProteoMiner Kit, each in combination with protein level fractionation via SDS PAGE or via offline C18 chromatography before trypsin digestion [168]. This study identified a total of 4385 proteins, 3064 (70%) of which were common between all methods. Another study reported 3400 proteins from pooled plasma samples from AD patients and healthy controls [169] by using a combination of immune depletion, HPLC fractionation, and 2D gel electrophoresis before LC-MS/MS analysis. Throughput loss resulting from extensive fractionation can be mitigated by the utilization of isobaric tags for relative and absolute quantification (iTRAQ) or tandem mass tags (TMT) for multiplexing analysis [170]. Keshishian et al. utilized abundant protein depletion, iTRAQ isobaric labeling, and offline basic PH reversed-phase fractionation to identify >4500 proteins in plasma with high reproducibility [171].

A recently developed nanoparticle-based method has reported promising results for the analysis of blood samples [172,173]. Over the last decade, there has been an increased understanding of the mechanisms of generation of ‘protein corona’, or the protein mixture bound at the interface between the surface of nanoparticles and a biological fluid, and which factors influence its composition [174,175]. Nanoparticles can reduce the proportion of highly abundant proteins and enrich low-abundance proteins in serum and plasma. Notably, the composition of protein coronas can be tuned for a swath of plasma proteins in a reproducible manner, enabling the use of nanoparticle-based automation workflows in combination with MS analysis, enabling protein coverage exceeding 2000 proteins [176]. Although initial studies utilizing nanoparticles suggest great promise for proteomic analysis of blood samples, these methods need to be further evaluated and improve reproducibility across different cores and institutions [177]. Improvements in standardization and automation will aid in these efforts [172].

### 4.2. MS-Based Approaches for the Discovery of Novel Therapeutic Targets

Proteomic methodologies can be leveraged for the discovery of novel senotherapeutic approaches in multiple ways. First, the identification of senescence-specific cell-surface proteins will unveil senotherapeutic targets. Second, proteomic profiling of protein–drug interactions associated with senolysis can aid in the identification of new pathways and proteins for the development of senolytics. As the protein targets of several existing senolytics are not yet known, the identification of protein interactors from existing senolytics presents an intriguing area for the discovery of novel senolytic pathways and the development of new drugs. Identifying specific drug/protein binding interactions may also enable the development of scientific interventions to address issues such as drug delivery, bioavailability, and off-target binding. Numerous methods for proteomic target discovery have surfaced over the last few decades, many of them relying on the quantitative unbiased measurements given by mass spectrometry.

Several existing methods, Thermal Proteome Profiling (TPP), Limited Proteolysis (LIP), and Stability of Proteins from Rates of Oxidation (SPROX), have proven to be effective in the successful identification and quantification of specific intracellular drug/target binding interactions. New approaches, in addition to these previously well-established methods, have also been recently introduced such as pulse proteolysis (PP) and Proteome Integral Solubility Alteration (PISA). Here, we provide a concise overview of both the newly emerging and previously well-established methodologies for proteomic target discovery.

#### 4.2.1. Limited Proteolysis Coupled to Mass Spectrometry (LiP-MS)

Lip-MS is a technique that compares proteolytic fragmentation patterns between a drug-treated group and a non-treated control [178]. Drug treatment induces conformational changes in drug-bound protein targets. Then, when a broad-spectrum protease is added to the samples for a short time, these conformational changes cause the protease to have altered accessibility to its cleavage sites at the site of drug–protein interactions. This results in distinct peptide fragment signatures in the drug-treated group compared to the control [178]. After disrupting the protein–drug interaction, specific digestion with trypsin is then used to further break down the remaining large peptide fragments, leaving tryptic peptides enriched at the sites of protein–drug interaction to be detected via MS. Lip-MS is useful in complex mixtures, such as cell lysates, for understanding the structure of proteins and any changes that occur due to drug binding [179].

#### 4.2.2. Pulse Proteolysis (PP)

Pulse proteolysis is another proteomic technique used for identifying potential drug targets, as well as protein–protein interactions. In this method, a cell lysate is incubated with a drug and urea, while a control sample is incubated under identical conditions without the drug. The drug will theoretically stabilize its target proteins and prevent their denaturation by urea. Then, a protease is introduced for a short period, referred to as a ‘pulse’. PP causes complete digestion of unfolded proteins, but drug-stabilized proteins remain intact [180]. After the pulse period, the reaction is quenched by rapidly lowering the temperature or by adding protease inhibitors. The protein is run on a gel, and the intact protein bands (higher molecular weight) are quantitatively compared. Drug-bound targets are indicated by an increased amount of intact protein in the drug-treated group [180]. A later improvement in this method incorporated mass spectrometry for the untargeted discovery of drug-binding interactions. In this variation, the portion of the gel containing intact protein is excised, digested in gel with trypsin, and analyzed via MS to compare the fraction of intact proteins between drug-treated and untreated groups [181].

#### 4.2.3. Stability of Proteins from Rates of Oxidation (SPROX)

Stability of Proteins from Rates of Oxidation (SPROX) employs a hydrogen peroxide buffer with increasing levels of a denaturant to generate chemical denaturation curves [182]. After an equal amount of time for the oxidation rate, the samples are quenched. Chemical denaturation curves are generated based on the oxidation of methionine residues on peptides detected and quantified with mass spectrometry [182]. SPROX reveals proteins that are stabilized or destabilized by a bound target molecule, indicated by either a right or left shift in the chemical denaturation curve [183]. It is especially beneficial in evaluating the disassociation constants of protein–ligand complexes as oxidation decreases protein stability. Since its first use, SPROX has been updated to allow for higher throughput detection with isobaric tags [183].

#### 4.2.4. Thermal Denaturation-Based Target Discovery: TPP, CETSA, and PISA

The Cellular Thermal Shift Assay (CETSA) compares thermal denaturation curves between drug-treated and control groups to identify potential drug targets. Drug-bound proteins exhibit stabilization that protects the drug–protein complex from thermal denaturation, allowing for the interaction to be identified by identifying proteins with a right-shifted curve [184]. Thermal Proteome Profiling (TPP) combines CETSA with mass spectrometry analysis to identify novel protein–drug interactions in untargeted mass spectrometry analysis [185]. Both techniques can be used in cell extracts and intact cells, enabling the identification of target engagement within the intact cellular environment. Notably, TPP can also be used to identify drug targets in tissues from animal studies. For example, one study demonstrated the dose-dependent engagement of methionine aminopeptidase-2 (MetAP2) with its inhibitor, TNP-470, in the livers of mice treated with the drug [184]. These techniques are beneficial for target discovery following treatment with a ligand, and TPP can also be applied to analyze downstream effects and other factors that affect protein stability such as post-translational modifications (PTMs) [185]. Finally, Proteome Integral Solubility Alteration (PISA) pairs thermal denaturation with the use of isobaric labeling to enable sample multiplexing during MS analysis [186]. By combining the soluble protein fractions obtained after a thermal gradient, targets can be identified by looking at the area under the curve across plexes within each sample [186]. PISA also allows for comparative quantification at a 10-or-more-fold increased throughput when compared to CETSA/TPP.

#### 4.2.5. Size-Exclusion Chromatography and Affinity Selection Mass Spectrometry (SEC-ASMS)

SEC-ASMS is a hybrid analytical technique that combines the size separation capability of SEC with further separation via affinity selection which is then analyzed with MS [187,188]. SEC is a liquid chromatography method that separates molecules based on size and shape. Affinity selection allows for the separation of ligand-bound proteins in a complex mixture. After the SEC separation, the eluted protein fractions are subjected to affinity selection, adding the ligand of interest, to selectively capture and enrich their target molecules from the complex protein mixture [187,188]. After affinity selection, captured protein–ligand complexes are analyzed using mass spectrometry to characterize the interaction and identify targets [187,188]. SEC-ASMS is also helpful for examining protein complexes, protein–protein interactions, and mapping post-translational modifications to study their function.

#### 4.2.6. Targeting and Quantifying Senescent Cells through the ‘Surfaceome’

For decades there have been efforts to identify and quantity proteins on cell surfaces because of their value as markers of cell identity and as therapeutic targets. A prime example of this is immunophenotyping, a method that essentially exploits the presence or absence of specific cell-surface markers to count and discriminate cell populations using antibody detection. Immunophenotyping is now a tool of choice to probe the phenotype and function of the immune system [189]. This method can be used to count and discriminate specific lymphocyte populations, including progenitor cells and various hematopoietic lineages, with ever-increasing granularity based on the presence or absence of cell-surface markers [189,190]. Knowing the identity and quantity of proteins on the cell surface has enormous potential benefits for translational research and the development of disease diagnostics and therapeutics, not only for its established benefit in immunophenotyping, but for the potential to identify therapeutic drugs, immunotherapy targets, and cell-surface targets for the isolation of cell populations.

Mass spectrometry-based proteomics has been applied to a variety of senescence models to identify surface markers that can be used to target these cells for senotherapy (Table 1). For example, unbiased mass spectrometry-based studies have identified dipeptidyl peptidase 4 (DPP4) to be enriched on the surface of senescent versus proliferating fibroblasts [191]. Importantly, DPP4 can be exploited for both sorting senescent cells via flow cytometry and for targeting them in vitro using antibody-dependent cell-mediated cytotoxicity. A subsequent study demonstrated that senescence-associated DPP4 activity can be pharmacologically targeted to improve hemostasis and plaque stability in old mice [192]. Another promising example of a therapeutically relevant surface marker is the urokinase-type plasminogen activator receptor (uPAR), originally identified as a preferentially expressed marker on the surface of senescent human lung adenocarcinoma cells and melanocytes [193]. In the same study, the authors demonstrated that chimeric antigen receptor (CAR) T cells that target uPAR efficiently eliminate senescent cells in vivo and extend the survival of mice with lung adenocarcinoma following treatment with MEK and CDK4/6 inhibitors, and ameliorated liver fibrosis in two independent models [193]. These examples highlight the utility of cell-surface proteins as potential senotherapeutic targets. For a comprehensive description of surfaceome markers of senescence, we refer readers to recent reviews [11,194].

Despite its huge benefits in diagnostics and cell phenotyping, immunophenotyping is a limited tool for the discovery of new cell-surface markers. First, it is limited to the set of cell-surface proteins that are already known and have high-quality antibodies. Secondly, immunophenotyping has limited multiplexing capability. Thus, unbiased approaches are critically needed for the discovery of new surfaceome targets that will aid in the discrimination of the highly diverse and heterogeneous cell types in the human body. Technological advancements in mass spectrometry approaches now enable comprehensive and unbiased identification and quantification of surfaceome proteins. Over the past decade, an increasing number of sample preparation workflows, computational tools, and curated surfaceomes have emerged to aid in the discovery and validation of cell-surface proteins using mass spectrometry, and are reviewed more extensively elsewhere [200,201]. The most rigorous methods for identification of the cell-surface proteome and the surface interactome utilize biorthogonal chemistry to biotinylate cell-surface proteins for subsequent affinity enrichment. For example, the cell-surface capture (CSC) [202] method utilizes non-cell-permeable reagents to biotinylate cell-surface glycoproteins, which comprise 90% of cell-surface proteins. Following the labeling of live cells in culture, labeled glycoproteins can be enriched either at the protein or peptide levels before mass spectrometry analysis. Modifications of CSC include Cys-Glyco-CSC, which allows the co-enrichment of peptides that are linked to the glycopeptides via disulfide bonds, and Lys-CSC, which allows labeling and enrichment of all surface-exposed lysine-containing proteins [203]. Modern variations of CSC have enhanced its utility for smaller sample inputs [204] and implemented automation, such as autoCSC [205,206] and µCSC [207], resulting in vastly improved sensitivity, reproducibility, and throughput and making surfaceome characterizations of many samples and cell types more feasible. Another recent innovation termed LUX-MS enables a precise spatiotemporal elucidation of the surface interactome of ligands or antibodies using a light-mediated reaction, thus enabling the precise mapping of cell-surface nanoscale organization with targeted therapies [208].

A critical aspect of mass spectrometry-based identification of the surfaceome is the correct annotation of surface proteins from the huge proteomic dataset, which requires a stringent bioinformatic analysis based on specific characteristics of surface proteins. A widely used approach is gene ontology (GO) annotations to identify which proteins are reported to be associated with the cell surface or plasma membrane. More rigorous filtering criteria for the identification of surface glycoproteins after CSC rely on the detection of a mass shift of 0.984 Da that corresponds to the deamidation of asparagine residues through enzymatic deglycosylation during the enrichment process. The selected peptides can be further filtered for the presence of the conserved NXS/T motif that is present in more than 90% of the N-glycosylation sites, allowing the exclusion of the peptides that might have co-eluted during the enrichment process [205]. Similarly, Lys-CSC requires the selection of all peptides that contain a mass shift of 145.019 Da corresponding to the 3-(carbamidomethylthio) propanoyl modification on the lysine residues [203].

Numerous software tools and databases can be employed to aid in the identification, filtering, and prioritization of candidate cell-surface proteins identified from mass spectrometry-based approaches. A suite of tools in the CellSurfer portal [207] (https://www.cellsurfer.net/ (accessed on 29 September 2023)), developed by the Gundry laboratory, is one such resource that aids in annotating and scoring candidate cell-surface markers from mass spectrometry data. The site includes several tools [207,209,210] that enable the rigorous bioinformatic validation, prioritization, ranking, and visualization of candidate surface proteins for future validation and clinical studies. Protter (https://wlab.ethz.ch/protter/start/ (accessed on 29 September 2023)) is another online open-source tool that allows visualization of peptides in the protein topology and provides evidence to validate whether the identified peptides are exposed on the cell surface [211].

Increased interest in exploring the cell surfaceome as a source of potential biomarkers has led to the development of online databases that can serve as a reference for future surfaceome studies. The Cell Surface Protein Atlas [212] (https://wlab.ethz.ch/cspa/ (accessed on 29 September 2023)) is a repository of mass spectrometry-derived cell-surface proteins from 41 human and 31 mouse cell types, providing a snapshot of the surface proteome from different cell types, including cancer cells. Similarly, the in silico human surfaceome (http://wlab.ethz.ch/surfaceome/ accessed on 29 September 2023)) is an online database containing 2886 proteins predicted using a machine-learning-based predictor, SURFY, developed by the Wollschied laboratory [213]. Despite these available databases, the unique cell surfaceome signature has not yet been described for many cell types and disease states, and thus continued efforts and innovation in surface protein discovery and annotation are needed.

There are also potential limitations to utilizing the surfaceome for targeting. First, just as in other markers, cell-surface markers are heterogeneous on senescent cells. Even in presumably homogenous cell culture conditions, not all induced senescent cells express surface markers such as DPP4 [191]. This is particularly true for markers of senescence, since senescent cell populations are inherently heterogenous and markers identified via bulk proteomic approaches may overestimate the ability of a protein to be a true marker of senescence. Strategies coupling single-cell technologies with surface labeling can provide a better approach to identifying markers from such heterogenous populations [200,201]. Secondly, it is not yet known how well surface markers distinguish different populations of senescent cells based on their phenotypes. Another technical challenge is that the validation of surface markers discovered via MS is limited by the availability of highly sensitive and specific antibodies to cell-surface proteins. This may limit the validation of surfaceome candidates using orthogonal approaches such as flow cytometry.

## 5. Conclusions and Future Directions

The applications of mass spectrometry-based proteomic technologies and workflows explored in this review can significantly accelerate the development of clinically relevant biomarkers and therapeutic targets to treat senescence-associated pathologies. Biomarkers and therapeutic targets are critical to the effective management and treatment of the senescence burden. Protein biomarkers of senescence, though temporal and heterogeneous by cell type, have proven to be clinically relevant through the prediction of age-related clinical outcomes such as mortality, multimorbidity, loss of mobility, loss of strength, and others. These studies, and further exploratory studies of the associations between senescence burden and specific diseases and pathologies of aging, will be valuable for informing the outcomes and populations that should be targeted with senotherapeutics. While there are examples of successful therapies targeting specific markers, the field is too young to know exactly which senescent cells are the most appropriate markers for targeting, and whether ‘bad’ senescent cell populations can be discriminated from neutral or good ones via a panel of markers. The comprehensive profiling of proteins associated with senescent cells in different contexts (tissue, inducer, and duration), and the validation of their impacts on age-related pathologies, will be key in identifying and discriminating specific subsets of senescent cells. Although great progress has been made in profiling senescence-associated proteins, validated and specific proteomic biomarkers are still lacking for the translation of senotherapeutics in clinics. Since the accumulation of senescent cells, the SASP, and the complexity of the process continues to grow, proteomic technologies will continue to be critical in profiling senotype-specific (specific to the senescence cell sub-type) biomarkers of aging and senescent cell burden, generating new potential drug targets for senotype-specific drugs or immunotherapies, or to elucidate the molecular mechanism of cellular senescence.

Validation of the presence of senescence-associated biomarkers in biofluids, such as plasma, urine, and CSF, among others, and the development of clinical biomarkers are important future steps for the field. However, due to the large dynamic range of protein concentrations, these tissues pose a challenge for proteomic analysis. With the recent explosion of proteomic technologies to measure large numbers of proteins in plasma and serum, quite a few options are currently available to those who are interested in performing biomarker discovery studies. We anticipate the increased use of these technologies in aging and senescence studies will be critical for biomarker discovery and validation in human biofluids. Additionally, the application of mass spectrometry-based proteomic workflows that now enable the identification and quantification of PTMs and proteoforms, such as DIA and top-down proteomic methods, will increase the specificity and likely the predictive power of biomarkers. Ultimately, the application of emerging proteomic technologies has great potential to impact the field of clinical geroscience and the treatment of senescence-associated pathologies in humans.

## Figures and Tables

**Figure 1 biology-12-01301-f001:**
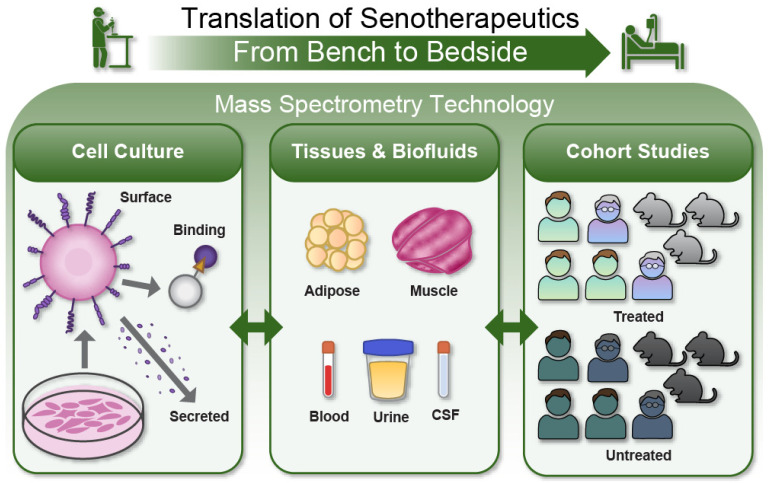
A translational pipeline for senotherapeutic development is accelerated by mass spectrometry technology. Developing senotherapeutics to be more clinically applicable necessitates a robust research pipeline that operates between in vitro models, tissue and biofluid samples, and larger cohort studies. Findings from each step build upon one another, whether applying a basic biological finding to a more complex system or examining a patient’s response to treatment at a more fundamental level. In this paradigm, mass spectrometry technology has helped bridge the bench-to-bedside gap with its unique precision, specificity, and high throughput. We envision how this technology can continue enabling novel discoveries at every stage of this translational pipeline to advance treatments for aging-related diseases.

**Figure 2 biology-12-01301-f002:**
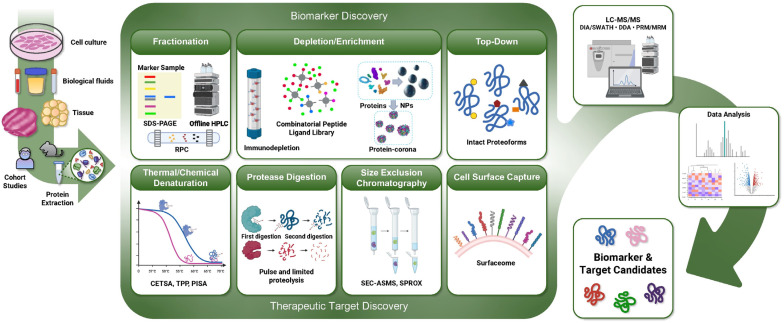
Emerging proteomics workflows for senotherapeutic biomarker and target discovery: Proteins extracted from cells, biological fluids, or tissues are evaluated using LC-MS/MS methodologies followed by data analysis. Various proteomic sample preparation workflows aid the discovery of senescence-associated biomarkers with greater depth and specificity (top panel). Depletion, enrichment, and fractionation techniques reduce the complexity of samples such as blood, enabling proteome profiling with greater depth. Additionally, top-down approaches enable identification and quantification at the proteoform-level to improve specificity. Another set of proteomic workflows can be leveraged for the identification of senotherapeutic targets (bottom panel). These workflows identify specific senescence-assocaited proteins for targeting, such as cell-surface proteins or proteins that interact with senolytic compounds.

**Table 1 biology-12-01301-t001:** Surface markers from senescent cells identified using mass spectrometry-based proteomics. RS = replicative senescence; OIS = oncogene-induced senescence; IR = irradiation; PM = plasma membrane; ADCs = antibody–drug conjugates; ADCC = antibody-dependent cell-mediated toxicity.

Surface Protein	Cell/Tissue Type	Senescence Inducer	Strategy	Senolytic Validation	Reference
DEP1/PTPRJ/CD148	Bladder cancer (EJp21 and EJp16); human lung fibroblast (IMR-90); human fibrosarcoma (HT1080p21); mouse lung adenomas (^V600E^BRAF); human melanocytes	p21/p16 overexpression; RS, RAS OIS - - -	PM isolation, in-gel digestion, and LC-MS/MS		[195]
B2MG	Bladder cancer (EJp21 and EJp16); mouse lung adenomas (^V600E^BRAF)	p21/p16 overexpression -	PM isolation, in-gel digestion, and LC-MS/MS	Gold nanoparticles, ADCs	[13,195,196]
NOTCH1	Human lung fibroblast (IMR-90); mouse pancreatic neoplasm (p48-cre)	HRAS^G12V^, OIS, Etoposide Kras^G12D^ OIS	SILAC		[197]
DPP4/CD26	Human lung fibroblast (WI-38, IMR-90); human aortic endothelial cells (HAEC); human umbilical vein endothelial cells (HUVEC); mouse embryonic fibroblasts (MEF)	RS, IR, Doxorubicin IR IR HRAS^G12V^ OIS	PM isolation, in-gel digestion, and LC-MS/MS	ADCC	[191]
SCAMP4	Human lung fibroblast (WI-38, IMR-90); human aortic endothelial cells (HAEC); human umbilical vein endothelial cells (HUVEC)	RS, IR, Doxorubicin, HRAS^G12V^ OIS IR IR	PM isolation, in-gel digestion, and LC-MS/MS		[198]
CD24	Bone marrow mesenchymal cells from INK-ATTAC mice	-	CyTOF		[199]

## Data Availability

Not applicable.

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
