# Peer review of "Translating Senotherapeutic Interventions into the Clinic with Emerging Proteomic Technologies"

_biology, 2023, doi:10.3390/biology12101301_

Round 1
Reviewer 1 Report
1. Within the special issue “Cellular senescence in health and disease”, this review focuses on proteomic approaches to acquisition of knowledge in senescence biology, with special emphasis on advances in biomarker discovery for senolytic therapy strategies. This topic is highly relevant and fits well into the special issue.
2. Given their expertise in technologies for identifying and validating protein biomarkers, the authors have provided a comprehensive review on the application of recent proteomic methodologies for profiling senotype-specific biomarkers of senescent cells, with the ultimate goal of generating novel potential drug targets. The senior author has contributed significantly to this field.
3. The article is relatively well structured and well written. After the introductory remarks, the manuscript starts with a few examples of animal and clinical studies that have demonstrated the relevance of anti-oxidant, anti-inflammatory (anti-SASP) agents, or of modulation of the PI3K and PKB/Akt, or P53/P21 pathways. This is followed by a section on SASP, correlations of SASP levels in plasma or in tissues/biofluids with frailty, performance score, and results in clinical trials. This part has provided the appropriate background for the present review.
4. In section 4, the authors have provided a concise yet detailed overview of methodologies on proteomic approaches for SASP biomarkers such as limited Proteolysis Coupled to Mass Spectrometry (LiP-MS), Pulse Proteolysis (PP), Stability of Proteins from Rates of Oxidation (SPROX), Size Exclusion Chromatography and Affinity Selection Mass Spectrometry (SEC-ASMS), Targeting and Quantifying Senescent Cells through the ‘Surfaceome’. I think this is an excellent overview on novel proteomic technologies for the discovery of novel markers and targets and may eventually facilitate the detection of low-abundance proteins, specific proteoforms, and identification of senescence specific cell surface proteins for possible seno-therapeutic drug targeting.
These are the strengths of the article.
With the ultimate goal of translating the knowledge gained from lab bench to bedside, there are a number of issues that might enhance the quality of this article:
1. Do we know what are our target cells? Do we have a clear definition of senescent cells in the context of aging? Thus far, the definition of “markers” or “constellation of markers” for senescent cells varied widely, dependent on organs and tissues and also on the differentiation stages in the developmental trajectory. Will advances in proteomic technologies facilitate the identification of mechanisms upstream and downstream of elevated SASP in aging of specific tissues.
2. Even if the respective target cells (senescent cells) are identified for a specific tissue, do we know if we hit the right target. For example, it took us decades to realize that we were able to achieve short-lived remissions of acute myeloid leukemia by targeting proliferating or more differentiated leukemia cells but the leukemia initiating cells, or leukemia stem cells, stay intact. Subsequently the disease recurs. The theme of this review is very focused on SASP and the respective factors associated with SASP. Is it possible that they just represent downstream targets, (like leukemia cells) whereas the cells initiating the senescence process remain intact? Should we also discuss alternative approaches that target the senescence initiating cells?
3. the authors have deliberated extensively on the merits of surface markers both for diagnostic as well as targeted strategies. Just in analogy to immunophenotyping, the markers are not as specific as we wished them to be. Some discussion on the pitfalls and limitations might be appropriate.
Not applicable
Reviewer 2 Report
Please see document attached

Minor editing of English language required
Round 2
Reviewer 1 Report
The authors have addressed the issues raised satisfactorily.